# Stability Engineering of Recombinant Secretory IgA

**DOI:** 10.3390/ijms25136856

**Published:** 2024-06-22

**Authors:** Kathrin Göritzer, Richard Strasser, Julian K.-C. Ma

**Affiliations:** 1Department of Applied Genetics and Cell Biology, University of Natural Resources and Life Sciences, 1190 Vienna, Austria; richard.strasser@boku.ac.at; 2Institute for Infection and Immunity, St. George’s University of London, London SW17 0RE, UK; jma@sgul.ac.uk

**Keywords:** secretory immunoglobulin A, IgA, antibody engineering, monoclonal antibody, plant molecular pharming, aerosolization, protein assembly, protein stability

## Abstract

Secretory IgA (SIgA) presents a promising avenue for mucosal immunotherapy yet faces challenges in expression, purification, and stability. IgA exists in two primary isotypes, IgA1 and IgA2, with IgA2 further subdivided into two common allotypes: IgA2m(1) and IgA2m(2). The major differences between IgA1 and IgA2 are located in the hinge region, with IgA1 featuring a 13-amino acid elongation that includes up to six *O*-glycosylation sites. Furthermore, the IgA2m(1) allotype lacks a covalent disulfide bond between heavy and light chains, which is present in IgA1 and IgA2m(2). While IgA1 demonstrates superior epitope binding and pathogen neutralization, IgA2 exhibits enhanced effector functions and stability against mucosal bacterial degradation. However, the noncovalent linkage in the IgA2m(1) allotype raises production and stability challenges. The introduction of distinct single mutations aims to facilitate an alternate disulfide bond formation to mitigate these challenges. We compare four different IgA2 versions with IgA1 to further develop secretory IgA antibodies against SARS-CoV-2 for topical delivery to mucosal surfaces. Our results indicate significantly improved expression levels and assembly efficacy of SIgA2 (P221R) in *Nicotiana benthamiana*. Moreover, engineered SIgA2 displays heightened thermal stability under physiological as well as acidic conditions and can be aerosolized using a mesh nebulizer. In summary, our study elucidates the benefits of stability-enhancing mutations in overcoming hurdles associated with SIgA expression and stability.

## 1. Introduction

Most of the monoclonal antibodies being approved or in clinical trials are of the IgG isotype [1,2]. Production technologies and purification protocols are well established, and regulatory agencies are acquainted with relevant safety issues. However, new antibody formats such as IgA are gaining more and more interest [3,4,5]. Out of all antibody classes, the most abundant isoform in the human body is IgA [6]. Three molecular forms of IgA exist: monomeric, dimeric and secretory IgA, each with distinct characteristics and functions in the immune system. Secretory IgA constitutes a first line of mucosal defense against invading pathogens. It is a heavily glycosylated multimeric protein consisting of two monomeric IgA molecules covalently linked by the joining J-chain (JC) and the secretory component (SC), which wraps around the antibody complex and confers resistance to proteolytic degradation, along with protection in low pH environments [7]. Being adapted to mucosal secretions means SIgA has an intrinsic advantage as a potential therapeutic targeting mucosal pathogens compared to other antibody classes. In airway infections, SIgA has been shown to neutralize influenza viruses and prevent virus-induced pathology in the upper respiratory tract, performing at times better than IgG [8,9]. This is likely to be due to its ability to bind antigens with high avidity on the mucosal surface, interact with mucins via the SC and prevent adherence to the epithelium, which is a process called immune exclusion [7,10].

Passive immunization would involve the topical administration of pathogen-specific SIgA directly to the oral, nasal, respiratory or gastro-intestinal mucosa to enhance the protective mucosal barrier. The respiratory application of SIgA therapy was highlighted against COVID-19 with monoclonal SIgAs showing higher neutralization against SARS-CoV-2 compared to IgG antibodies [11,12]. With functional efficacy and applications against a wide range of mucosal pathogens, it is understandable why SIgA emerges as an attractive candidate in the growing field of antibody therapy. However, several factors must be addressed before SIgA fulfills its therapeutic potential. An important step would be to identify an optimal IgA constant region scaffold on which to build SIgA antibodies with different specificities.

In nature, immunoglobulin A exists in several isotypes, namely IgA1 and IgA2, of which the latter exists in two common allotypes (IgA2m(1), IgA2m(2)). The major differences between IgA1 and IgA2 are located in the hinge region, with IgA2 lacking the 13-amino acid elongation with up to six O-glycosylation sites in IgA1 (Figure 1). Furthermore, IgA1 has two N-glycosylation sites in the CH2 domain and the tailpiece. IgA2m(1) has additional N-glycosylation sites (Asn166 in the CH1 and Asn 337 in the CH2 domain). IgA2m(2) has an additional N-linked glycan at Asn211 in the CH1 domain [13,14]. These additional glycan sites create a greater proinflammatory response of neutrophils and macrophages [15]. The IgA2m(1) allotype lacks a covalent disulfide bond between heavy and light chains, which is present in IgA1 and in the IgA2m(2) allotype. Previous studies have shown that the presence of P221 in IgA2m(1) interferes with the formation of a heavy and light chain (HL) disulfide bond in the absence of C133. In this allotype, only heavy-chain dimer (H2) molecules are being efficiently formed with only small quantities of HL and H2L2 molecules present in secretions [16,17].

In recombinant antibodies, IgA1 showed better bivalent binding of special separated epitopes due to its extended flexible hinge region and increased neutralization of pathogens, while IgA2 was more effective in inducing effector functions and is more stable in terms of bacterial degradation at mucosal surfaces. This is believed to be due to the lack of the extended hinge region, which is susceptible to proteolytic degradation [15,18,19]. However, the unusual noncovalent linkage of the light chain and heavy chain pairing in the IgA2m(1) allotype may result in other production and stability concerns [14]. Stability needs to be addressed for SIgAs during expression, formulation and in terms of proteolytic degradation at the site of application. Regarding the latter, aerosolization is an attractive means for delivery to the upper respiratory tract, so conformational and thermal stability are important considerations. Engineering the IgA heavy chain to reduce sensitivity to bacterial proteases and environmental pH is also necessary. A potential strategy in antibody engineering is the introduction of covalent bonds in the form of disulfide bridges. For example, the introduction of a P221R mutation was previously demonstrated in CHO cell-produced IgA2m(1) to sterically allow the formation of a new disulfide bond between the heavy (Cys-220) and the light chain cysteines (Cys-214) [20].

The complexity of recombinant production of secretory IgA will drive the cost, which determines their viability as therapeutic products. Due to the need to transcribe and assemble four components, SIgA production is already a complex multi-step process which has proved challenging in different protein production systems. Attempts to produce SIgA in mammalian cells have resulted in only modest success [21,22,23]. Plants have emerged as an attractive platform for SIgA production, enabling complete assembly in planta without the need for in vitro processes. This method has achieved promising yields of up to 100 mg/kg of leaf fresh weight [12]. However, none of the available sequences are optimal either due to expression yields, low thermal stability, or high susceptibility to bacterial proteases. Here, we compare four different IgA2 versions with IgA1 to further develop secretory IgA antibodies against SARS-CoV-2 for topical delivery to mucosal surfaces.

## 2. Results

### 2.1. Recombinant Production and Characterization of Monomeric IgA Stability Mutants

Due to the previously reported low thermal stability of mIgA2m(1), as well as different SDS-PAGE mobility that suggested incomplete assembly, we performed site-directed mutagenesis of the IgA2m(1) allotype to generate three different mutants that would potentially enable the formation of disulfide bridges between the light chain (LC) and heavy chain (HC) of IgA2m(1) (Figure 1). Firstly, we introduced a mutation from aspartic acid (D) to cysteine (C) at position 133 that was shown to be responsible to form a disulfide bridge between the HC and the LC in the IgA1 allotype to generate mIgA2m(1)_D133C [24]. Secondly, we mutated the proline (P) adjacent to a cysteine residue that potentially forms a disulfide bridge between the HC and the LC close to the hinge region in the IgA2m(2) allotype to arginine (R) to generate mIgA2m(1)_P221R [17]. Lastly, we used the latter mutant to generate the less common IgA isotype, IgA2m(2), by further introducing serine (S) instead of proline (P) further upstream, resulting in an additional *N*-glycosylation site with the sequon NSS in the CH2 domain.

Variable heavy-chain (VH) domains from the SARS-CoV-2 mAbs were grafted on the native and mutated alpha chain constant region backbones. Variable light-chain (VL) domains were grafted onto the respective kappa light-chain (κLC) or lambda light-chain (λLC) constant region. All IgA variants were transiently expressed in glyco-engineered *Nicotiana benthamiana* ΔXT/FT plants [12,25]. Results are shown for COVA2-15, while 2E8 can be found in the Appendix A.

The immunoblotting of crude leaf extracts with IgA HC-specific antisera under non-reducing conditions showed a major band for COVA2-15 and 2E8 mIgA1 corresponding to the expected molecular weight of ~160 kDa (Appendix A). The predominant band for COVA2-15 mIgA2m(1) was ~100 kDa, which likely represents heavy-chain dimers. This band was also present, although to a much lesser extent in 2E8 λIgA2m(1), and it was barely visible for other λIgA2m(1) COVA1-22 and 2–15. In contrast, COVA2-15 mIgA2m(1)_P221R, mIgA2m(2) and mIgA2m(1)_D133C showed a weak single band at 160 kDa suggesting the full formation of disulfide bridges between the LC and HC. While this was also the case for 2E8 mIgA2m(1)_P221R, the expression levels of the other mutants was low and barely visible in immunoblotting.

The expression levels determined by ELISA of crude leaf extract showed the highest accumulation of monomeric IgA1 and IgA2m(1) after five to six days post-infiltration with 100–150 mg/kg leaf fresh weight (LFW) (Figure 2A), which is in accordance with previous reports [12]. While monomeric COVA2-15 IgA2m(1)_P221R displayed similar or even higher accumulation levels than parental IgA2m(1), the expression of mIgA2m(1)_D133C and mIgA2m(2) variants was substantially reduced.

Next, the transient production of monomeric IgAs was upscaled in *Nicotiana benthamiana* plants (200 g LFW). After affinity purification of clarified leaf extract, IgAs were subjected to size exclusion chromatography (Figure 2B). All COVA2-15 variants except COVA2-15 mIgA2m(1)_D133C displayed a major peak at the expected retention time for proteins with a mass of ~160 kDa. This corresponds to the monomeric structural unit with additional minor peaks at lower retention times representing high molecular weight aggregates (HMWAs). These HMWAs were significantly less abundant for mIgA2m(1)_P221R and mIgA2m(2). The overall purification yields after size-exclusion chromatography correlated to the accumulation levels determined by ELISA from crude extracts with high values reported for COVA2-15 mIgA1, IgA2m(1) as well as IgA2m(1)_P221R and low levels for the other mutants (Appendix A). Peaks representing mIgA were pooled and run on an SDS-PAGE under non-reducing conditions and visualized with Coomassie Blue (Figure 2C). As already seen with immune blotting of crude leaf extract, COVA2-15 mIgA1 variants display a major band with 160 kDa. On the other hand, COVA2-15 mIgA2m(1) shows a major band at 100 kDa, although a major peak at a retention time correlating with a molecular weight of 160 kDa was observed in SEC, and no degradation products of the heavy chain were observed on an SDS-PAGE under reducing conditions (Appendix A). This likely represents the full assembly of COVA2-15 mIgA2m(1) with noncovalent interaction of the LC and HC and the incomplete formation of a disulfide bridge between the LC and HC. This is further evidenced by the presence of a band at 45 kDa, likely representing LC dimers.

Finally, monomeric IgAs were tested for functionality in terms of binding to antigen RBD (Figure 2D). All mIgA variants displayed similar antigen-binding curves, resulting in similar EC_50_ values that are in accordance with previously reported values (Appendix A) [12]. Only the binding of COVA2-15 mIgA2(1)_D133C to RBD was slightly reduced.

The 2E8 mIgA variants showed similar expression levels to COVA2-15 with the highest accumulation reported for mIgA2m(1)_P221R. While the 2E8 IgA isotypes had similar SEC chromatograms to COVA2-15, they displayed different SDS-PAGE running behavior, with all variants showing a predominant band at 150 kDa including IgA2m(1), suggesting that disulfide bridges between the λLC and the HC are formed (Appendix A).

### 2.2. Recombinant Production and Characterization of Secretory IgA Stability Mutants

The mutant IgA2m(1)_P221R that showed the most promising expression levels and characteristics, in both COVA2-15 and 2E8 antibodies, was taken forward to be expressed as a secretory IgA (SIgA). Light and heavy chain pairs were co-expressed in the presence of the joining chain (JC) and secretory component (SC) to obtain fully assembled SIgA. The expression levels of fully assembled SIgA were 100 and 60 mg/kg LFW for COVA2-15 and 2E8, respectively. In both cases, the SIgA2m(1)_P221R variant expressed better than the SIgA2m(1) counterpart, which was equivalent to the SIgA1 version (Figure 3A and Appendix A). After affinity purification of crude leaf extract, SIgAs were subjected to size-exclusion chromatography (Figure 3B and Appendix A). COVA2-15 SIgAs displayed a major peak at early retention times (SIgA) and a minor peak at later retention times (mIgA) as previously reported [12].

Each of the eluted fractions was analyzed by ELISA to determine the ratio of fully functional and assembled secretory IgA as previously described. Recombinant SIgAs were captured with RBD and detected with anti-SC antibody and compared to total IgA, using an anti-IgA heavy chain antibody for capture and an anti-kappa or lambda light chain antibody for detection. Thereby, it was confirmed that the major peak of all variants contains fully assembled and functional SIgA as previously described [12]. The second minor peak represents monomeric IgA that was not assembled into polymeric IgA. The amount of free mIgA was reduced from around 20% to >10% in COVA2-15 SIgA2m(1)_P221R compared to SIgA1 and SIgA2m(1). For 2E8 SIgAs, that displays lower assembly efficacy into dimers (Appendix A), the amount of mIgA was ~35%, and this was increased for SIgA2m(1) and SIgA2m(1)_P221R to ~45%. Although the overall expression levels of fully assembled 2E8 SIgA2m(1)_P221R are increased compared to SIgA2m(1), the assembly efficacy is reduced.

Fractions containing fully assembled SIgA were pooled and run on SDS-PAGE. COVA2-15 SIgA variants displayed broad bands at 400 kDa and 800 kDa representing dimeric and tetrameric forms as previously reported for this antibody (Figure 3C) [12]. COVA2-15 SIgA2m(1) additionally displayed bands at 45 kDa representing LC dimers, which cannot be observed for COVA2-15 SIgA2m(1)_P221R. The 2E8 SIgAs displayed single bands at 400 kDa, suggesting the presence of mostly dimers with additional bands visible for SIgA2m(1)_P221R, which were probably caused by peak overlap and insufficient separation when samples were pooled (Appendix A). All SIgAs were further tested for antigen binding and displayed similar EC50 values (Figure 3D and Appendix A).

### 2.3. Increased Thermal Stability of IgA2m(1) Monomeric and Secretory IgA Mutants

To determine the influence of the alpha chain mutations on the thermal stability of the proteins, we performed differential scanning fluorimetry of all purified variants under physiological conditions (Figure 4A and Appendix A, Table 1 and Appendix A). The thermal unfolding of all IgAs followed a broad endotherm, which is monitored by changes in fluorescence as a function of temperature when fluorescent dye binds to hydrophobic regions of the protein as it unfolds. A very low initial fluorescence was observed for all variants except COVA2-15 mIgA2m(1)_D133C. This initial high fluorescence could indicate the improper folding and exposure of hydrophobic patches, as already suggested by the non-monodisperse peak during size-exclusion chromatography (Figure 2C). The thermal stability of the IgA mAbs was determined by calculating the midpoint temperature of unfolding. Generally, 2E8 λLC IgG and IgA variants were more stable than COVA2-15 κLC antibodies. While mIgA1 displays high thermal stability, mIgA2m(1) is significantly less stable with an average of 4 °C lower midpoint temperature of protein unfolding transition (T_m_). However, the stability mutants COVA2-15 mIgA2m(1)_P221R and mIgA2m(2) showed increased midpoint temperatures (>7 °C) compared to mIgA2m(1). This correlation was also observed for 2E8 variants although not to such an extent.

Next, we investigated the thermal unfolding of fully assembled SIgA. Thermal unfolding of SIgA was again represented by a broad endotherm (Figure 4B). The calculated midpoint temperature of SIgAs was similar to their monomeric counterpart with a significant increase in T_m_ for the SIgA2m(1)_P221R stability mutants compared to SIgA2m(1) for both COVA2-15 and 2E8 versions.

Lastly, we investigated the thermal stability of the SIgA isotypes in acidic conditions (Figure 4C and Appendix A), and the data are summarized in Table 2 and Appendix A. All variants were dialyzed against citric acid buffer (pH 3.5) and incubated for 1 h prior to thermal shift assays. Thermal unfolding of SIgAs at pH 3.5 was significantly shifted toward lower midpoint temperatures and was represented by a broad endotherm for SIgA2m(1), but two separate unfolding events were observed for SIgA1 and SIgA2m(1)_P221R variants. The first midpoint temperature likely represents unfolding of the CH2 and CH3 domain, while the second midpoint temperature represents the Fab domains [26]. When the P221R mutation is introduced, thermal stability of the Fab domains of the SIgA2 isotype is significantly improved in acidic conditions. Thermal shift assays after incubation in acidic conditions were not possible for IgG, because under the tested conditions, all IgG was degraded. Overall, introducing mutations to aid in the formation of disulfide bridges between the LC and the HC (IgA2m(1)_P221R and IgA2m(2)) increases thermal stability, particularly for COVA2-15, which has a kappa light chain.

### 2.4. Aerosolization of SIgA

Considering the potential delivery of SIgA mAbs to the upper and lower respiratory tracts through aerosols, all monomeric and polymeric IgA formats were aerosolized using the Omron MicroAir nebulizer, and the stability of the antibodies in the aerosolization condensate was characterized (Figure 5). When the IgAs were formulated in 1xPBS, aerosolization resulted in a significant loss of protein, with only 60–70% recovery for monomeric IgA1 and IgA2m(1) of the COVA2-15 variant, and 75% recovery for IgA2m(1)_P221R. Recovery was even less for SIgA (COVA2-15 SIgA1:37.2%, SIgA2: 32.9%) and was slightly increased to 46.6% for the IgA2m(1)_P221R mutant. Recovery levels were substantially improved for all antibodies when 0.05% Tween was added to the formulation (Figure 5A, Appendix A). Similar observations were made for 2E8 IgA (Appendix A).

The loss of monomeric and secretory antibody by aerosolization, when formulated in 1xPBS, was associated with low retained antigen-binding capacity for IgA1 and IgA2m(1) that was in the range of 55–65% for COVA2-15 and even less for 2E8 variants (Figure 5B and Appendix A). Interestingly, this loss of antigen-binding capacity was not associated with the formation of large amounts of aggregates in the condensate, as evidenced by size-exclusion chromatography (Appendix A). Surprisingly, introduction of the P221R mutation boosted the antigen-binding capacity of monomeric and SIgA2 in the aerosolization condensate from 50–60% to 100%.

The formation of subvisible aggregates and size distribution of COVA2-15 SIgA2m(1) and SIgA2m(1)_P221R before and after aerosolization was determined using dynamic light scattering (DLS; Figure 5C). The average size of 25 nm is in the expected range for dimeric and tetrameric SIgA. The polydispersity index was 0.165 for SIgA2m(1)_P221R and 0.214 for SIgA2m(1), suggesting some degree of multidispersity also represented in the broad band in SDS-PAGEs of SIgA and the presence of dimers (molecular weight 469 kDa) as well as tetramers (molecular weight 958 kDa, Appendix A). However, dynamic light scattering did not suggest a shift toward oligomers after aerosolization for SIgA2m(1)_P221R. This seemed to be the case for SIgA2m(1) when formulated in 1xPBS. Nonetheless, the total amount of HMWAs (100–350 nm) in the condensate was very low (volume < 0.3%) when formulated with 1xPBS or 1xPBS with the addition of 0.05% Tween, and it was generally even less for SIgA2m(1)_P221R.

## 3. Discussion

While there are many possible advantages of IgA in antibody therapy, there are several issues that need to be overcome. These include the efficiency of expression, production, purification, and complete assembly of recombinant IgA monoclonal antibodies with appropriate homogeneity as well as stability during topical delivery. In IgA2, while the shorter hinge may restrict the movements of the Fab regions to access antigens [27], it provides a functional advantage by being resistant to bacterial IgA1 proteases. This may explain why IgA2 is more abundant in the intestinal secretions where most of the bacteria reside. IgA1 may suffer stability issues in vivo, particularly at bacterially colonized sites. Additionally, *O*-glycosylation is typically diverse and difficult to control during biomolecule production, which limits regulatory and safety experience [28]. Importantly, if applied in serum, aberrantly hypogalactosylated natural IgA1 antibodies are critically involved in the development of IgA nephropathy, which is a common cause of renal failure [29].

Traditionally, IgA2m(1) has been described as being an antibody lacking covalent bonds between H and L, while IgA1 and IgA2m(2) have covalent HL disulfides. Previous studies have shown that this is not strictly true [16]. For IgA2m(1), some HL disulfide bonds do form, albeit only inefficiently, suggesting that in IgA2m(1), the cysteine residue involved in forming the HL disulfide is partially accessible, but the majority of the molecules fail to form this bond. Also, lambda light chain rather than kappa light chain is better at forming these disulfide bridges [20]. Additionally, in IgA2m(1), a significant amount of the noncovalent L chains are dimers, suggesting that the noncovalent L chains assume different orientations in IgA2m(1) and IgA2m(2). Thus, IgA2m(2) with covalently linked H and L chains may be more stable than IgA2m(1) in the milieu of the mucosal secretions with varying pH and salt concentrations.

We therefore decided to modify IgA2m(1), using two well-characterized SARS-CoV-2 specific antibodies, COVA2-15 and 2E8, that express well in plants [12]. We introduced a cysteine at position 133, which is found in the CH1 domain of IgA1 and is responsible for disulfide bonding with the LC. For another variant, a different mutation in the CH1 domain of the IgA2 heavy chain from proline to arginine at position 221 was made, which has already been described previously [16,20]. Finally, we introduced a further mutation in IgA2m(1)_P221R to introduce the *N*-glycosylation site at position Asn-212 in the CH1 to generate the common IgA2m(2) allotype.

The mutant IgA2m(2)_D133C was able to form disulfide bridges with the light chain, which was probably due to the proximity of C133 in the constant HC and C214 in the constant LC region [27,30]. However, expression in *N. benthamiana* plants was markedly reduced, and conformational and structural stability were significantly impaired. The mutant IgA2m(2)_P221R sterically enables the formation of an alternate disulfide bridge between the heavy and light chains in the IgA2m(1) allotype, which was similar to the IgA2m(2) allotype. Our results show that a single P221R amino acid exchange derived from the IgA2m(2) sequence is sufficient to prevent dissociation into heavy and light-chain homodimers in monomeric and secretory IgA. Introduction of the P221R mutation enhanced the expression levels of monomeric IgA reaching reported values of the respective IgG counterpart in plants (Figure 2) [12]. Furthermore, the mutation enhanced the stability of the protein and increased expression levels in plants. Surprisingly, adding an additional mutation to introduce the CH1 resident *N*-glycosylation site to generate mIgA2m(2) resulted in a significant reduction in expression in plants. A similar poorer expression of IgA2m(2) as well as assembly into multimeric IgA compared to other allotypes in plants has been reported previously [12,31,32,33].

The expression of multimeric SIgA is more complex than monomeric IgA, but it has been successfully reported for SIgA1 and SIgA2m(1) in plants [12]. Generally, a higher capacity for assembly into polymers in planta can be observed for IgA1 rather than IgA2m(1) [12,31]. The introduction of mutation P221R has positive effects on the assembly efficacy of kappa LC antibodies but not those with lambda light chains. However, the overall expression levels of fully assembled SIgA2m(1)_P221R increased compared to SIgA2m(1), reaching those of SIgA1, and the presence of free light chain dimers could not be detected.

In addition to achieving high accumulation levels, ensuring the stability of mAbs is paramount for the successful development of topical delivery systems in therapeutic applications [12,34]. Topical delivery to the respiratory tract through aerosolization, such as using the widely available Omron MicroAir nebulizer, has been explored [12]. However, the complex multimeric structure of antibodies, especially SIgA, adds further intricacy to the formulation process. Concerns regarding thermal denaturation during aerosolization highlight the importance of assessing thermal stability during preclinical development to ensure the viability of mAbs under different conditions. Prior studies have demonstrated that monomeric IgA1 and IgA2m(2) exhibit notable thermal stability with the unfolding of fully assembled secretory IgA remaining unexplored. However, the thermal unfolding of mIgA2m(1) is distinguished by a broad endotherm, signifying the onset of unfolding at temperatures approximately 6 °C lower than those observed for other allotypes. While glycosylation status is important for thermal stability, the formation of disulfide bridges between the L and H chain is even more critical [26]. This was reflected in the thermal stability determined here by differential scanning fluorometry, where κLC-mIgA2m(1) in particular showed a much reduced *T*_m_ compared to IgA1. With the introduction of further disulfide bonds between the L and H chain, thermal stability of not only monomeric but also secretory IgA2 improved drastically.

The topical administration of mAbs via oral delivery also holds potential. Challenges include the susceptibility of mAbs to degradation by gastric acid and proteases [35]. Higher robustness in acidic conditions combined with the reports of higher resistance of SIgA against gastric enzymes like pepsin due to its unique structure highlights why SIgA is well suited for mucosal applications [36]. This was also observed here, where under highly acidic conditions, SIgA1 and particularly stability engineered SIgA2_P221R displayed much improved midpoint temperatures of unfolding compared to IgG mAbs.

The increased thermal stability of engineered IgA2m(1) also translated into a better recovery of fully functional IgA after aerosolization. When formulated in 1xPBS, only 40% of the total protein was recovered. This was associated with a significant loss of protein and/or activity except for stability engineered monomeric and secretory IgA2m(1)_P221R. However, no aggregates were detectable in the condensate. The loss could be mostly reversed by adding 0.05% Tween-20 (Polysorbate-20) to the antibody preparation. While formulation seems to be the key determinant for successful aerosolization, stabilizing mutations make IgA more robust for topical delivery using less complex formulations.

In conclusion, while IgA holds promise for monoclonal antibody therapy, overcoming challenges in expression, purification, and stability is imperative. The distinctive structural features of IgA, such as its hinge region and glycosylation patterns, present both advantages and complexities in therapeutic development. Our study highlights the importance of understanding the structural nuances of different IgA allotypes, such as IgA2m(1) and IgA2m(2), in order to engineer antibodies with improved stability and functionality. Through targeted mutations, such as the P221R substitution, we have demonstrated enhanced stability and assembly efficacy, particularly in the context of secretory IgA. Furthermore, our results underscore the importance of formulation optimization in facilitating successful aerosolization for topical delivery with stabilizing mutations proving to be instrumental in enhancing the resilience of IgA antibodies under varying conditions. Ultimately, our study contributes to advancing the understanding and development of IgA-based therapeutics for mucosal applications, offering promising avenues for combating infectious diseases and other mucosal disorders.

## 4. Materials and Methods

### 4.1. Construct Design and Cloning

Heavy-chain sequences of human gamma-1 (AAA02914.1), alpha-1 (AAT74070.1) or alpha-2m(1) (AAT74071.1) constant regions were cloned together with a human Ig heavy chain leader sequence (‘MDMRVPAQLLGLLLLWLPGARC’) separated by a BsaI type II restriction site into pDONR-based plasmids and have been previously described [12]. Similarly, constant human lambda and kappa light chain were cloned together with the human light chain leader sequence (‘MDMRVPAQLLGLLLLWLPGARC’) also separated by a BsaI type II restriction site into pDONR. Site-directed mutagenesis of the pDONR-alpha-2m(1) scaffold to generate pDONR-alpha-2m(1)_D133C, pDONR-alpha-2m(1)_P221R and pDONR-alpha-2m(2) was performed with the primers described in Appendix A using the QuickChange kit II XL Site-Directed Mutagenesis Kit (Agilent, Santa lara, CA, USA). *Nicotiana benthamiana* codon-optimized sequences for the heavy and light-chain variable regions of COVA2-15 (QKQ15273.1, QKQ15189.1), COVA1-22 (QKQ15169.1, QKQ15253.1), 2-15 (PDB: 7L57_H, 7L57_L) and 2E8 IgG mAbs were costume synthesized by GeneArt (Thermo Fisher Scientific, Waltham, MA, USA) and flanked with BsaI type II restriction sites as previously described [12,37,38]. Using Golden Gate assembly, the variable heavy-chain sequences were cloned into the pDONR-based heavy chain scaffold plasmids. Variable light-chain fragments of COVA2-15 were inserted into human kappa constant region pDONR scaffolds (AAA58989.1) and COVA1-22, 2-15 and 2E8 were inserted into lambda constant region pDONR scaffolds (CAA40940.1) [23]. Full-length heavy and light-chain genes were separately subcloned into the binary high expression vector pEAQ-HT-DEST3 using gateway cloning [39]. Human secretory component (SC) and joining chain (JC) constructs cloned separately into pEAQ-HT have been described previously [23]. The pEAQ-HT plant expression vectors containing the gamma and alpha heavy chains as well as the kappa and lambda light chains were transformed into *Agrobacterium tumefaciens* strain GV3101 (Leibniz Institut DSMZ-Deutsche Sammlung von Mikroorganismen und Zellkulturen GmbH, Braunschweig, Germany, DSM 12364) by electroporation.

The construct for expression of the receptor-binding domain (RBD) of the SARS-CoV-2 spike (PDB: 6VYB, R319-F541) with a C-terminal 6xHis-tag cloned into pCAGGS mammalian expression vector has been described previously [12].

### 4.2. Transient Expression of IgG and IgA Variants in N. benthamiana

Agrobacterium strains carrying the relevant constructs were cultured overnight at 28 °C in Lysogeny Broth (LB) supplemented with 25 µg/mL rifampicin and 50 µg/mL kanamycin. For the expression of IgG or monomeric IgA1 and IgA2 variants, the overnight cultures containing the respective constructs for the heavy and light chains were diluted in infiltration buffer (10 mM MES, 10 mM MgSO_4_, and 0.1 mM acetosyringone) to achieve an optical density at 600 nm (OD_600_) of 0.1. For secretory IgA variants, the heavy and light-chain constructs were diluted to an OD_600_ of 0.05. They were then mixed with the joining chain construct at an OD_600_ of 0.2 and the secretory component construct at an OD_600_ of 0.1. Subsequently, the Agrobacterium solutions were introduced into 6–8-week-old glycoengineered *Nicotiana benthamiana* ΔXT/FT that are almost completely deficient in β1,2-xylosylation and core α1,3-fucosylation, resulting in glycoproteins carrying human-like *N*-glycosylation, as previously described, by vacuum infiltration [25,40]. The plants were cultivated in a controlled environment room at 25 °C under a 16/8 h light/dark cycle. After 5 days, the infiltrated leaf material was harvested, and crude leaf extract was prepared by blending with three volumes of ice-cold phosphate-buffered saline (PBS) pH 7.4 containing 0.1% (*v*/*v*) Tween 20. The homogenized leaf material was filtered through a Miracloth filter (MilliporeSigma, Burlington, MA, USA) and centrifuged at 20,000× *g* for 1 h, which was followed by filtration through 0.45 µm pore-size filters (Durapore membrane filter, MilliporeSigma, Burlington, MA, USA).

### 4.3. Purification of IgG and IgA Variants from Crude Leaf Extract

The clarified leaf extracts underwent purification using columns packed with either Pierce™ Protein A resin for the isolation of IgG and COVA2-15 IgA variants or a CaptureSelect™ IgA affinity matrix (both from Thermo Fisher Scientific, Waltham, MA, USA) for the purification of 2E8 IgA variants, which were pre-equilibrated with PBS. Proteins were eluted using 0.1 M glycine at pH 2.7, which was followed immediately by the addition of 10% (*v*/*v*) 1 M Tris-HCl at pH 9.0 to neutralize the pH. Fractions containing the protein of interest were combined and dialyzed against PBS at 4 °C overnight using a dialysis cassette with a molecular weight cut-off (MWCO) of 10 kDa (Slide-A-Lyzer, Thermo Fisher Scientific, Waltham, MA, USA). Subsequently, the pooled and dialyzed protein fractions were concentrated using Amicon centrifugal filters with an MWCO of 100 kDa (MilliporeSigma, Burlington, MA, USA) and subjected to size-exclusion chromatography (SEC) on a HiLoad 16/600 Superdex 200 pg column (GE Healthcare, Chicago, IL, USA) pre-equilibrated with PBS at pH 7.4. The SEC was performed using an ÄKTA pure FPLC system (GE Healthcare, Chicago, IL, USA).

### 4.4. ELISA

For the quantification of IgA mAbs in clarified crude extract of infiltrated N. benthamiana plants, ELISA plates were coated with 250 ng/well goat polyclonal antibody to human anti-alpha chain (ab97211, Abcam, Cambridge, UK) in PBS pH 7.4 at 4 °C overnight. After blocking with PBS containing 2% (*w*/*v*) BSA and 0.1% Tween 20 (*v*/*v*), clarified crude plant extracts were added to the wells in normalized concentrations and incubated for 1.5 h at 37 °C. As standards, purified human IgA (P80-102, Bethyl Laboratory, Montgomery, TX, USA) and IgA from human colostrum (I2363, Sigma, USA) were used. The detection of secretory IgA variants was carried out with a mouse anti-secretory component antibody (SAB4200787, Sigma, USA), which was followed by an HRP-labeled anti-mouse antibody (SAB5300168, Sigma, USA). For monomeric IgA variants, HRP-labeled anti-kappa (A18853, Invitrogen, Waltham, MA, USA) or anti-lambda light-chain (ab200966, Abcam, Cambridge, UK) antisera were used. After incubation for 1 h at 37 °C, plates were developed using TMB (Thermo Fisher, USA) substrate, the reaction was stopped with 2 M H_2_SO_4_ and the read-out was performed on an Infinite F200 Pro plate reader (Tecan, Männedorf, CH, Switzerland) at 450 nm.

The ratio of functional and fully assembled SIgA to total IgA in each size-exclusion fraction was determined by similar ELISA assays. Capture was with 100 ng/well purified recombinant RBD-His. Purified mAbs were diluted to 2 µg/mL in blocking solution, added to RBD-coated plates in normalized concentrations and incubated for 1.5 h at 37 °C. The detection of secretory component or antibody kappa or lambda chains was carried out as described above.

To determine the binding of the purified recombinant mAbs to SARS-CoV-2 RBD, ELISA plates were coated with 100 ng/well purified RBD-His. The purified mAbs were added to the wells in normalized concentration. For detection, HRP-labeled anti-human kappa or lambda light chain antibodies were used as above. Half-maximal concentration (EC_50_) was calculated in GraphPad Prism 9.0.

### 4.5. SDS-PAGE

First, 5 µg of purified mAbs was resolved on a NuPage 4–12% Bis/Tris gel (Life Technologies, Carlsbad, CA, USA) and stained with InstantBlue (Expedeon, Harston, UK).

### 4.6. Immunoblotting

Diluted crude leaf extracts were resolved on a NuPage 4–12% Bis/Tris gel (Life Technologies, UK) and then blotted on nitrocellulose membrane by semi-dry transfer, and bands were visualized using HRP-labeled anti-alpha HC antibody (ab97215, Abcam, Cambridge, UK).

### 4.7. Differential Scanning Fluorimetry (DSF)

Differential scanning fluorimetry (DSF) was conducted using a CFX real-time PCR instrument (Bio-Rad Laboratories, Hercules, CA, USA) in 1×PBS buffer at pH 7.4. Each sample was analyzed in triplicate, utilizing 96-well MicroAmp Fast reaction plates with 25 µL of sample per well. Monoclonal antibodies (mAbs) were diluted to a concentration of 1 mg/mL in the formulation buffer. SYPRO Orange Fluorescent Dye (Thermo Fisher Scientific, Waltham, MA, USA) was diluted 1000-fold from a 5000× *g* concentrated stock to prepare the working dye solution in the formulation buffer before addition to the mAb samples. Thermal denaturation was initiated by gradually increasing the temperature from 25 to 95 °C at a rate of 0.05 °C/s. Fluorescence intensity measurements were recorded using the FRET channel. Automated data processing of thermal denaturation curves involved truncating the dataset to eliminate post-peak quenching effects. The first derivative approach to calculate T_m_ was used. In this method, T_m_ is the temperature corresponding to the maximum value of the first derivative of the DSF melting curve.

### 4.8. Aerosolization of Monoclonal Antibodies

COVA2-15 and 2E8 SIgA1 were aerosolized using a commercially available Omron Micro Air U22 electronic mesh nebulizer (Omron Healthcare, Milton Keynes, UK) as previously described [41].

### 4.9. Dynamic Light Scattering

DLS measurements were performed as described previously with protein concentrations of 500 µg/mL in 1×PBS pH 7.4 supplemented with 0.05% Tween on a Malvern Zetasizer nano-ZS (Malvern Panalytical, Malvern, UK) in a 12 mL quartz cuvette [12]. Samples were measured at 25.0 °C, and the LS was detected at 173° and collected in automatic mode. The mean values and SEs of the number weighted diameter were calculated from three measurements for each sample, and each reported value is an average.

### 4.10. SEC-LS

SEC-LS was used to characterize the recombinant expressed proteins in solutions relating to their purity, native oligomers or aggregates, and molecular weights as previously described [12]. Analyses were performed on an OMNISEC multidetector gel permeation chromatography (GPC)/SEC system equipped with a refractive index detector, a right-angle LS detector, a low-angle LS detector and a UV/visible light photodiode array detector (Malvern Panalytical, Malvern, UK). A Superdex 200 Increase 10/300 GL column (Cytiva, Marlborough, MA, USA) was used and equilibrated with Dulbecco’s PBS without Ca and Mg, P04-361000 (PAN-Biotech, Germany), as running buffer. Experiments were performed at a flow rate of 0.5 mL min^−1^ at 25 °C and analyzed using OMNISEC software version 11.40 (Malvern Panalytical, Malvern, UK). Proper performance of the instrument was ensured by calibration and verification using the 200 mg Pierce BSA standard (Thermo Fisher Scientific). Before analysis, samples were centrifuged (16,000× *g*, 10 min) and filtered through 0.2 mm Durapore PVDF centrifugal filter(s) (MilliporeSigma, Burlington, MA, USA). A 100 mL volume of each sample was injected, having different concentrations between 0.1 and 0.5 mg/mL.

## Figures and Tables

**Figure 1 ijms-25-06856-f001:**
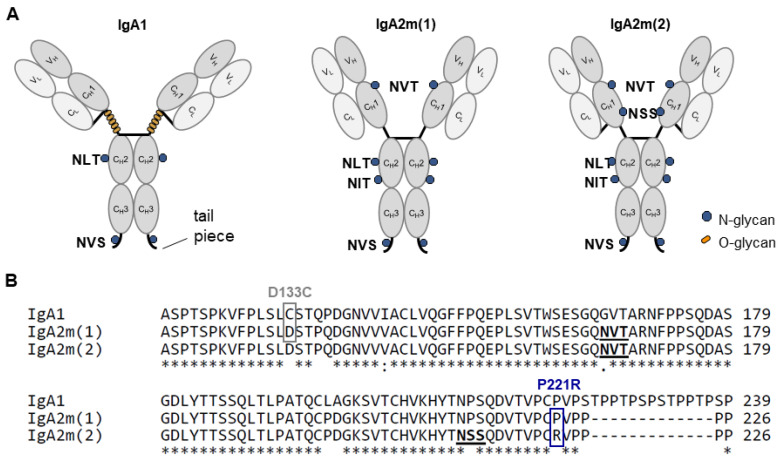
Illustration of the monomeric structural units of IgA isotypes. (**A**) Schematic illustration of the structure and glycosylation sites of the IgA isotypes IgA1, IgA2m(1) and IgA2m(2). The light chain is colored in light gray, and the heavy chain is colored in dark gray. N-glycans found in the different isotypes are indicated by blue dots. The O-glycans in the elongated hinge region of IgA1 are marked by orange dots. Below the protein sequence is a key denoting conserved (*), conservative mutations (:), semi-conservative mutations (.) and non-conservative mutations ( ). (**B**) Sequence overlay of human IgA1, IgA2m(1) and IgA2m(2) CHI domain including the hinge region. *N*-glycosylation sites are in bold and underlined. Targeted regions for site-directed mutagenesis have been indicated by grey (D133C) and blue (P221R) boxes.

**Figure 2 ijms-25-06856-f002:**
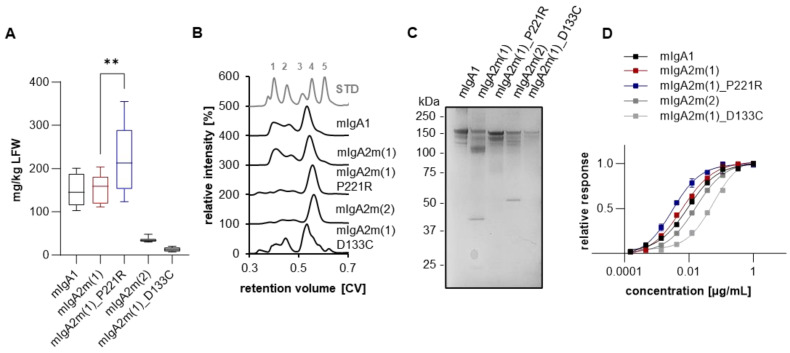
Expression and characterization of stability engineered monomeric COVA2-15 IgA variants. (**A**) Monomeric IgA1 and IgA2 recognizing the SARS-CoV-2 spike protein were transiently expressed in *Nicotiana benthamiana* ΔXT/FT plants. Expression levels were quantified by sandwich ELISA in crude leaf extracts. Detection of monomeric IgA was with HRP-labeled anti-kappa (COVA2-15) or anti-lambda light-chain (2E8) antibodies. Quantification data represent the mean of two technical repeats of three independent infiltrations of 3 plants each ± SD. One-way ANOVA was performed to compare the groups; ** *p* < 0.01. (**B**) Normalized size-exclusion chromatograms of affinity-purified monomeric IgA isotypes from infiltrated *N. benthamiana* ΔXT/FT leaves. Curves are representatives of two individual runs with similar results. Values were normalized based on the highest signal of each chromatogram. As a protein molecular weight standard, the Gel Filtration Markers Kit (Sigma Aldrich, St. Louis, MO, USA) was used (gray line; 1: Thyroglobulin 669 kDa, 2: Apoferritin 443 kDa, 3: β-Amylase 200 kDa, 4: Alcohol Dehydrogenase 150 kDa, 5: Albumin 66 kDa). (**C**) SDS-PAGE under non-reducing conditions of affinity and size-exclusion purified plant-produced monomeric IgA1/IgA2 visualized by Coomassie Brilliant Blue staining. (**D**) Determination of EC_50_ values of monomeric IgA variants to the receptor-binding domain (RBD) of the SARS-CoV-2 spike protein. Each value is the mean ± SD from three independent measurements.

**Figure 3 ijms-25-06856-f003:**
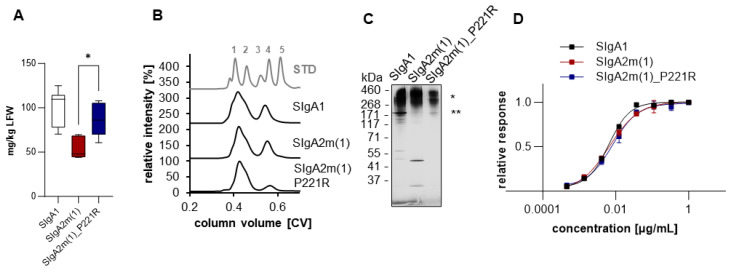
Expression and characterization of stability engineered COVA2-15 SIgA. (**A**) Secretory IgA1 and IgA2 recognizing the SARS-CoV-2 spike protein were transiently expressed in *N. benthamiana* ΔXT/FT plants. Expression levels were quantified by sandwich ELISA in crude leaf extracts. Secretory IgA antibodies were detected using anti-secretory component antibodies. Quantification data represent the mean of two technical repeats of three independent infiltrations of 3 plants each ± SD. One-way ANOVA was performed to compare the groups; * *p* < 0.03. (**B**) Normalized size-exclusion chromatograms of affinity-purified monomeric IgA isotypes from infiltrated *N. benthamiana* ΔXT/FT leaves. Curves are representatives of two individual runs with similar results. Values were normalized based on the highest signal of each chromatogram. As a protein molecular weight standard, the Gel Filtration Markers Kit (Sigma Aldrich) was used (gray line; 1: Thyroglobulin 669 kDa, 2: Apoferritin 443 kDa, 3: β-Amylase 200 kDa, 4: Alcohol Dehydrogenase 150 kDa, 5: Albumin 66 kDa). (**C**) SDS-PAGE under non-reducing conditions of affinity and size-exclusion purified plant-produced secretory IgA1/IgA2 visualized by Coomassie Brilliant Blue staining; fully assembled SIgA (*), monomeric IgA (**). (**D**) Determination of EC_50_ values of secretory IgA variants to the receptor-binding domain (RBD) of the SARS-CoV-2 spike protein. Each value is the mean ± SD from three independent measurements.

**Figure 4 ijms-25-06856-f004:**
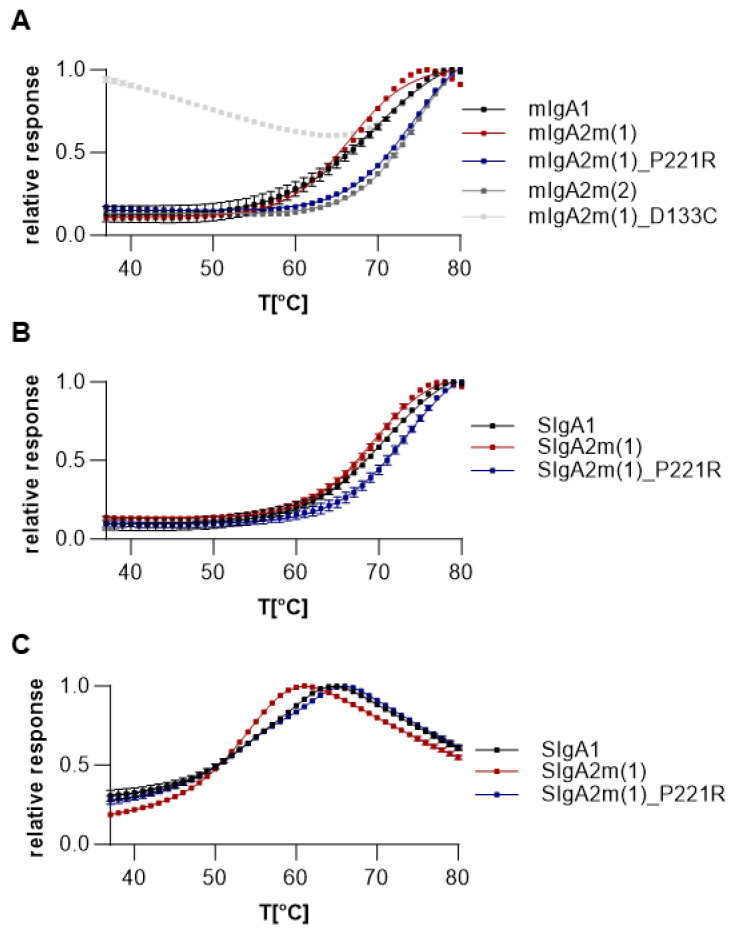
Thermal unfolding of COVA2-15 monomeric and secretory IgA. Differential scanning fluorimetry curves of monomeric (**A**) and secretory (**B**) IgA isotypes in 1xPBS buffer pH 7.4 or citrate buffer pH 3.5 (**C**). Experiments were performed at 1 mg/mL. DSF curves are the mean of three technical repeats of one out of three independent experiments with similar outcome.

**Figure 5 ijms-25-06856-f005:**
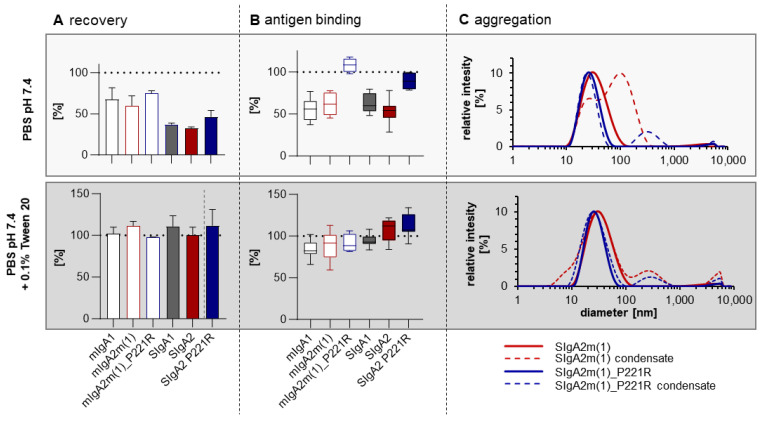
Aerosolization of COVA2-IgA antibodies using the Omron MicroAir U22 portable mesh nebulizer. First, 1 mL of monoclonal antibody with a concentration of 500 µg/mL was nebulized, and condensate was collected. (**A**) Protein concentration in the condensate after aerosolization was measured using absorbance at 280 nm. (**B**) Antigen binding of samples before and after nebulization were tested by ELISA with equal protein loading. Values for recovery and antigen-binding activity are means ± SD of at least three independent aerosolization experiments. (**C**) Dynamic light scattering of COVA2-15 SIgA2m(1) and SIgA2m(1)_P221R before and after aerosolization using the Omron MicroAir U22 portable mesh nebulizer. The mean diameter and the homogeneity were measured using a Malvern Zetasizer nano-ZS (Malvern Instruments Ltd., Worcestershire, UK) at 25 °C. Each sample was measured in triplicate.

**Table 1 ijms-25-06856-t001:** Midpoint temperatures of thermal unfolding (T_m_) of COVA2-15 monomeric and secretory IgA mutants at physiological conditions. Thermal shift assays were performed in 1xPBS buffer pH 7.4. Values represent the median of at least three independent differential scanning fluorimetry experiments with 3 repeats each ± SD; n.d, not determined.

	COVA2-15
mIgA1	71.4 ± 2.2
mIgA2	66.1 ± 0.7
mIgA2m(1)_P221R	74.2 ± 0.3
mIgA2m(2)	74.4 ± 0.3
mIgA2m(1)_D133C	n.d
SIgA1	71.2 ± 0.7
SIgA2m(1)	69.6 ± 1.4
SIgA2m(1)_P221R	73.8 ± 0.4

**Table 2 ijms-25-06856-t002:** Midpoint temperatures of thermal unfolding (T_m_) of COVA2-15 secretory IgA mutants in acidic conditions. Thermal shift assays were performed in citrate buffer pH 3.5. Values represent the median of three independent differential scanning fluorimetry experiments with 3 repeats each ± SD.

	COVA2-15	
	T_m1_ [°C]	T_m2_ [°C]
SIgA1	53.1 ± 0.6	60.0 ± 0.0
SIgA2m(1)	54.3 ± 0.2	/
SIgA2m(1)_P221R	53.3 ± 0.2	61.9 ± 0.3

## Data Availability

Any additional information required to reanalyze the data reported in this paper is available from the lead contact upon request.

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
