# Peer review of "Stability Engineering of Recombinant Secretory IgA"

_ijms, 2024, doi:10.3390/ijms25136856_

Round 1

Reviewer 1 Report

Comments and Suggestions for Authors

Göritzer and collaborators describe the design of mutants to increase recombinant SIgA stability against SARS-CoV-2. The SIgA protein is a relevant molecule with great possibilities for application in mucosal immunotherapy to defend against pathogens. The manuscript is a fascinating work and is very well structured, however it is necessary to address some aspects:

Figure S1: change “SDS-APGE” to “SDS-PAGE”

Line 146. The calibration graph of the molecular exclusion column should be included.

Line 157. A monodisperse peak represents a sample of a single type of molecule; how is this data from Figure 2B reaffirmed? since the peaks are not representatives of a monodisperse sample?

Line 223. In Figure 3C, a band of 800 kDa is not; the maximum marker is 400 kDa. How did you determine the presence of 800 kDa proteins (tetrameric form)?

To determine thermal stability due to temperature denaturation, a native (folded) and a denatured (unfolded) population must be considered in the calculations. In Figure 4, not all endotherms present in the displayed population can be determined, so a Tm cannot be determined; in these cases, not even a trend in the curve of the displayed region is observed.

It is necessary to compare the protein expression data in comparison with those published by other groups where the same expression system is used.

In the electrophoresis method, only gel 4-12% acrylamide was used. The complete image of the gel must then be shown to determine whether or not there are other major oligomeric states, as in Figure S4.

It is of utmost importance to address the observations about the experiments. This will not only help in revising the conclusions but also enhance the accuracy and completeness of your work. Please understand that this feedback is meant to improve your manuscript, not to criticize it.

Comments on the Quality of English Language

Minor language revision is required.

Author Response

Answers to Reviewer Nr. 1

Göritzer and collaborators describe the design of mutants to increase recombinant SIgA stability against SARS-CoV-2. The SIgA protein is a relevant molecule with great possibilities for application in mucosal immunotherapy to defend against pathogens. The manuscript is a fascinating work and is very well structured, however it is necessary to address some aspects:

Figure S1: change “SDS-APGE” to “SDS-PAGE”

Thank you, we have changed the spelling accordingly.

Line 146. The calibration graph of the molecular exclusion column should be included.

We have included the protein molecular weight standard size-exclusion chromatography run in figures 2B and 3B.

Line 157. A monodisperse peak represents a sample of a single type of molecule; how is this data from Figure 2B reaffirmed? since the peaks are not representatives of a monodisperse sample?

We have assumed monodispersity of the major peak based on the retention time corresponding to the protein standard run on the same column as well as bands identified by SDS-PAGE under non-reducing conditions. We have changed the phrasing accordingly and described the observation in line 157.

Line 223. In Figure 3C, a band of 800 kDa is not; the maximum marker is 400 kDa. How did you determine the presence of 800 kDa proteins (tetrameric form)?

From a previous study that was cited in line 223 (Göritzer et al 2024) we determined the molecular weight of COVA2-15 SIgA by SEC coupled to multi-angled light scattering that determines the molecular weight of protein label free in solution. There it was shown that COVA2-15 SIgAs occur mostly as tetramers with a molecular weight of around 800 kDa. This observation has been cited accordingly.

To determine thermal stability due to temperature denaturation, a native (folded) and a denatured (unfolded) population must be considered in the calculations. In Figure 4, not all endotherms present in the displayed population can be determined, so a Tm cannot be determined; in these cases, not even a trend in the curve of the displayed region is observed.

The Tm values were calculated using a standardized and well reported approach based on the first derivative of the thermograms which are based on the maximum change of the endotherm. This was possible for all variants due to very low initial fluorescence, which means most of the protein is folded and in native state at room temperature. This, however was not the case for mIgA2m(1)_D133C only, for which no Tm value could be calculated (stated in the Table 1).

It is necessary to compare the protein expression data in comparison with those published by other groups where the same expression system is used.

Thank you for the suggestion. We have previously published expression of fully assembled SIgA variants and will now reference the expression levels reported there. Other previous studies have not consistently reported expression and purification yields of fully assembled SIgA. We have made the changes in line 136.

In the electrophoresis method, only gel 4-12% acrylamide was used. The complete image of the gel must then be shown to determine whether or not there are other major oligomeric states, as in Figure S4.

In all figures of non-reducing SDS-PAGES we have displayed the full acrylamide gel. Pictures were cut just below the wells. All original uncut SDS-PAGE figures have been provided to the journal.

It is of utmost importance to address the observations about the experiments. This will not only help in revising the conclusions but also enhance the accuracy and completeness of your work. Please understand that this feedback is meant to improve your manuscript, not to criticize it.

We thank the reviewer for their helpful feedback. 

Reviewer 2 Report

Comments and Suggestions for Authors

In this manuscript, Göritzer and colleagues aim to address important challenges in the production and application of IgA class antibodies, exemplifying their antibody engineering strategy with IgA antibody variants recognising the SARS-CoV-2 spike protein.

They compare four different IgA2 versions with IgA1 to further develop secretory IgA antibodies against SARS-CoV-2 in plant cells for topical delivery to mucosal surfaces. They demonstrate improved expression levels and assembly efficacy of SIgA2 (P221R). An engineered SIgA2 displays heightened thermal stability under physiological and acidic conditions and can be aerosolized using a mesh nebulizer. Overall, the study is well conducted and of significant interest to the immunology and antibody engineering and therapy field. The findings highlight the possibility of stability-enhancing mutations in overcoming some key hurdles associated with SIgA generation, study and application. Suggestions:

1.     Abstract: The following statement is unclear because there is no preceding explanation of the different IgA isotypes and forms: “IgA exists in distinct isotypes with varied functionalities. While IgA1 demonstrates superior epitope binding and pathogen neutralization, IgA2 exhibits enhanced effector functions and stability against mucosal bacterial degradation.” Please revise the abstract to make this clearer to a non-expert/antibody specialist audience.

2.     Figure 1 is well presented. Since a significant strength of this work lies with the novel engineering of IgA antibodies, a schematic detailing the cloning approach is appropriate and will enhance the manuscript.

3.     Figures 2A and 3A: if the authors claim that some forms are produced in higher titres it would be important to provide statistical evaluations of these.

4.     Figures 2B and 3B: The size exclusion chromatography data indicate less pure monomeric product than is commonly seen with IgG antibodies. Please explain this and in Figure 3B: provide quantitative assessments of the proportion of monomeric antibody and minor peak products.

5.     Figures 2C and 3C: Bands on SDS PAGE gels appear to be on various concentrations. Are these generated by loading the same amount of protein for each sample? Please provide a clear statement and explanation.

6.     Figures 2D and 3D: Add a table of EC50 values to accompany each Figure 2D and 3D and Include error bars for each binding curve.

7.     Figure 4: Provide error bars for the technical replicates of the representative experiment.

8.     Figure 5: What do the error bars represent in Figure 5A and 5B? How many independent experiments?

9.     Apart from target binding and EC50 measurement, a potency assessment such as neutralization could be employed to compare the functionality of the different variants and will point to the translational relevance of the engineered antibodies.

10.  Very minor corrections are required throughout the document, e.g., lines 62-63, 87, 88, 96, 290, 346 to correct grammar, remove double spaces.

11.  A revision of the referencing style consistency is required, since some references are placed following the sentence full stops, while others towards the second half of the manuscript are displayed in superscript form.

Author Response

Answers to Reviewer Nr. 2

In this manuscript, Göritzer and colleagues aim to address important challenges in the production and application of IgA class antibodies, exemplifying their antibody engineering strategy with IgA antibody variants recognising the SARS-CoV-2 spike protein.

They compare four different IgA2 versions with IgA1 to further develop secretory IgA antibodies against SARS-CoV-2 in plant cells for topical delivery to mucosal surfaces. They demonstrate improved expression levels and assembly efficacy of SIgA2 (P221R). An engineered SIgA2 displays heightened thermal stability under physiological and acidic conditions and can be aerosolized using a mesh nebulizer. Overall, the study is well conducted and of significant interest to the immunology and antibody engineering and therapy field. The findings highlight the possibility of stability-enhancing mutations in overcoming some key hurdles associated with SIgA generation, study and application. Suggestions:

  1. Abstract: The following statement is unclear because there is no preceding explanation of the different IgA isotypes and forms: “IgA exists in distinct isotypes with varied functionalities. While IgA1 demonstrates superior epitope binding and pathogen neutralization, IgA2 exhibits enhanced effector functions and stability against mucosal bacterial degradation.” Please revise the abstract to make this clearer to a non-expert/antibody specialist audience.

We thank the reviewer for their suggestion. We have modified the abstract to enhance clarity.

  1. Figure 1 is well presented. Since a significant strength of this work lies with the novel engineering of IgA antibodies, a schematic detailing the cloning approach is appropriate and will enhance the manuscript.

Thank you for your comment. We have modified Figure 1 to more clearly illustrate the cloning strategy.

  1. Figures 2A and 3A: if the authors claim that some forms are produced in higher titres it would be important to provide statistical evaluations of these.

Thank you. We have added statistical analysis for Figures 2A and 3A, as well as for the corresponding supplemental Figures S2 and S4.

  1. Figures 2B and 3B: The size exclusion chromatography data indicate less pure monomeric product than is commonly seen with IgG antibodies. Please explain this and in Figure 3B: provide quantitative assessments of the proportion of monomeric antibody and minor peak products.

The additional peaks represent high molecular weight (HMW) aggregates, which can also be observed in IgG produced in plants. The relative amount of HMW aggregates can vary from batch to batch, depending on the quality and method of extraction and the affinity purification step, but can account for up to 30%.

  1. Figures 2C and 3C: Bands on SDS PAGE gels appear to be on various concentrations. Are these generated by loading the same amount of protein for each sample? Please provide a clear statement and explanation.

Generally, we load 5 µg of protein on SDS-PAGE gels. Variations in intensities can result from differences in contrast and brightness when scanning the gel.

  1. Figures 2D and 3D: Add a table of EC50 values to accompany each Figure 2D and 3D and Include error bars for each binding curve.

EC50 values of all IgA variants are provided in the supplemental materials (Tables S2 and S3). Both figures include error bars for each binding curve; however, the curves were highly reproducible, resulting in very minor error bars that are barely visible for many of the data points.

  1. Figure 4: Provide error bars for the technical replicates of the representative experiment.

All figures now include error bars. However, the technical replicates are often nearly identical, so the error bars are not clearly visible.

  1. Figure 5: What do the error bars represent in Figure 5A and 5B? How many independent experiments?

The bars represent the mean of three independent aerosolization experiments ± SD, as described in the legend of Figure 5.

  1. Apart from target binding and EC50 measurement, a potency assessment such as neutralization could be employed to compare the functionality of the different variants and will point to the translational relevance of the engineered antibodies.

Previous investigations have already explored virus neutralization of COVA2-15 and 2E8 SIgA1 and SIgA2. Given that we made no significant alterations to the structure of the antigen binding regions of the antibodies, and that antigen binding was demonstrated to be unaltered, substantial changes in neutralization capacity are not expected.

  1. Very minor corrections are required throughout the document, e.g., lines 62-63, 87, 88, 96, 290, 346 to correct grammar, remove double spaces.

Thank you, we have adjusted accordingly.

  1. A revision of the referencing style consistency is required, since some references are placed following the sentence full stops, while others towards the second half of the manuscript are displayed in superscript form.

The discrepancies in references likely occurred during the formatting process by the journal. I have carefully revised the references, and I am hopeful that this issue will be rectified by the journal.

Reviewer 3 Report

Comments and Suggestions for Authors

Review of "Stability engineering of recombinant secretory IgA" by Kathrin Göritzer, Richard Strasser and Julian Ma in Journal of Molecular Sciences

The authors produced SARS-CoV-2 IgA antibodies in planta and describe the purification process. They compare different allotypes of IgA2 as well as introduce two different mutations. 

Major issues:
Even though the production, purification and in vitro characterization of sIgA2 is interesting, the manuscript lacks a proper application. The authors claim that the mutation P221R leads to increased stability and better recovery after aerosolization. However, they never show this effect in vivo. They repeatedly cite their own article Göritzer et al, Molecular Therapy (2024) where they did perform mouse studies with sIgA1 aerosols showing that they have the setup to prove their claims. A similar animal study with sIgA2m(1)_P221R is necessary.

Minor issues:

Nowhere in the manuscripts do the authors mention which spike variant the original antibodies were raised against and which RBD variant they used to test the binding. Given the fast mutation rate of SARS-CoV-2, especially in the RBD, this is vital information.

Fig1A: The figure legends states that IgA2 has a elongated hinge region. This is wrong.

Introduction line 98: The authors quote their own study saying that the in planta expression outperforms mammalian expression systems. However, they do not investigate mammalian expression systems in the quoted article.

Fig3C: the gel is overloaded with protein, single bands are not visible. Needs to be improved.

Discussion line 339: IgA2m(1) does not have covalent HL disulfide bridges.

Author Response

Answers Reviewer Nr. 3

The authors produced SARS-CoV-2 IgA antibodies in planta and describe the purification process. They compare different allotypes of IgA2 as well as introduce two different mutations.

Major issues:

Even though the production, purification and in vitro characterization of sIgA2 is interesting, the manuscript lacks a proper application. The authors claim that the mutation P221R leads to increased stability and better recovery after aerosolization. However, they never show this effect in vivo. They repeatedly cite their own article Göritzer et al, Molecular Therapy (2024) where they did perform mouse studies with sIgA1 aerosols showing that they have the setup to prove their claims. A similar animal study with sIgA2m(1)_P221R is necessary.

This study is indeed a follow on from the 2024 Molecular Therapy paper. However, repeating the murine experiments would have been costly, and not necessarily helpful for the following reason. The engineered antibodies are human SIgAs, and although we used the mouse model to show proof of concept for mucosal protection, the mucosal environment of the upper respiratory tract in mice is different from humans in important ways. For example, the composition of the mucous layer is different, as is the microbiome and therefore the array of proteases that the SIgA would be exposed to.

For the purposes of this study, we felt it sufficient to demonstrate in vitro equivalence P221R and the non-mutated antibody, and to demonstrate activity or superiority of the P221R mutant our preference is to perform these studies in humans, which is our next objective for SIgA antibodies.

Minor issues:

Nowhere in the manuscripts do the authors mention which spike variant the original antibodies were raised against and which RBD variant they used to test the binding. Given the fast mutation rate of SARS-CoV-2, especially in the RBD, this is vital information.

The original antibodies and the RBD used in this study have been described in detail in Göritzer et al. 2024 and referenced. These antibodies, developed in 2021, were raised against the Wuhan strain and the RBD also derives from this strain. COVA2-15 does not bind to current Omicron variants. Nevertheless, this study serves as a proof of principle, focusing on the SIgA backbone rather than the specific target or variable region. Adjustments would be necessary in further development of SIgA for therapeutic use to address the evolving nature of the virus.

Fig1A: The figure legends states that IgA2 has a elongated hinge region. This is wrong.

Thank you for pointing this out, we corrected the spelling mistake.

Introduction line 98: The authors quote their own study saying that the in planta expression outperforms mammalian expression systems. However, they do not investigate mammalian expression systems in the quoted article.

Indeed, we did not investigate mammalian expression systems. However, there have been several studies attempting to express fully assembled SIgA in mammalian expression systems, albeit with modest success. We have referenced these studies accordingly and changed the text for more clarity.

Fig3C: the gel is overloaded with protein, single bands are not visible. Needs to be improved.

Unlike simpler monomeric antibodies like IgG that can be detected as single clear bands on SDS-PAGE gels, SIgA almost invariably appears as a broad smear on SDS-PAGE gels due to its various oligomeric states and extensive glycosylation. Reducing loading amounts only results in smears with lower intensity, as illustrated in Göritzer et al. 2024 and Teh et al. 2022.

Discussion line 339: IgA2m(1) does not have covalent HL disulfide bridges.

Thank you, we corrected the spelling mistake.

Round 2

Reviewer 1 Report

Comments and Suggestions for Authors

Thanks for addressing the comments. The research is fascinating and has a significant contribution IgA antibodies.

Comments on the Quality of English Language

The manuscript requires a minimum language revision, which can be addressed during editing before publication.

Author Response

The manuscript has been reviewed by the senior author Prof. Julian Ma, who is a native English speaker.

Line 28: Removed comma in line 28

Line 33: changed sentence structure for more clarity

Line 49: Corrected “COVD-19” to “COVID-19

Line 51: changed sentence structure for more clarity

Line 65: “an” to “a”

Line 83: “non-covalent” to “noncovalent”

Line 88: removed “in order”

Line 90: “Disulphide” to “Disulfide”

Line 94: added comma

Line 96: “4” to “four”

Line 102: added comma

Line 161: “expect” to “except”

Line 234: “that” to “which”

Line 244: “as” to “when”

Line 247: added comma

Line 269, 272, 278: changed sentence structure for more clarity

Line 279: added comma

Line 281: “as” to “because”

Line 282: : changed sentence structure for more clarity

Line 296: removed space

Line 356: removed comma

Line 358: “Cysteine” to “cysteine”